# Antagonistic Evolution for LLM Tool Use

## Abstract

Tool use has emerged as a pivotal mechanism for enhancing Large Language Models (LLMs), allowing them to interact with external tools to solve complex tasks and access knowledge beyond their static pre-trained parameters. However, most existing studies rely on advanced LLMs to improve tool-use capabilities via data synthesis, often resulting in suboptimal data quality or mismatched task difficulty, thereby limiting model performance. To address these limitations, we propose a novel antagonistic evolution framework for tool-use tasks, involving a query-generation model and a tool-use model updated in an adversarial manner. The query-generation model is optimized to produce increasingly challenging and high-quality queries, which the tool-use model then learns to solve. This adversarial process is iteratively executed, enabling both models to co-evolve and progressively enhance the tool-use capabilities. Experiments on three comprehensive tool-use benchmarks demonstrate evolving performance improvements, validating the effectiveness of our approach.

## 1 Introduction

Large Language Models (LLMs) are typically pre-trained on giga-token corpora using large-scale GPU clusters, and are subsequently served in a static manner. The prohibitive training cost prevents them from being frequently updated with new knowledge or adapted to dynamic environments. To enable LLMs to tackle complex problems, particularly in interactive settings, enhancing their tool-use ability has become a central research goal in the AI community. This direction has also been emphasized in state-of-the-art foundation models, such as GPT-5[1], Claude[2], Kimi-K2 (Team et al., 2025a), and LongCat (Team et al., 2025b), thereby advancing the development of agentic AI.

Existing approaches to improving LLMs' tool-use ability primarily rely on synthesizing labeled training data for post-training. Typically, these methods construct automated pipelines that prompt advanced models (e.g., GPT-4 or Claude-3.5) to generate queries and solutions, thereby simulating multi-turn interactions among users, assistants, and tools (Liu et al., 2025; Zhang et al., 2025). While such approaches can efficiently produce large-scale tool-use datasets at relatively low cost, their effectiveness is often constrained by data quality and target-model adaptivity, as highlighted in prior work (Liu et al., 2025). On the one hand, even state-of-the-art LLMs may introduce hallucinations when generating synthetic data, leading to quality deficiencies (Chen et al., 2024a). On the other hand, training on data that are either overly simplistic or excessively difficult may fail to benefit, or even harm, the target model's performance by causing negative transfer and disrupting its original knowledge structure (Wang et al., 2019). Inspired by recent advances in *self-play* paradigms for LLMs, where models iteratively generate and refine training data to achieve self-improvement, one promising direction is to adaptively tailor synthetic data to better match the target model's evolving capability. Such methods demonstrate that models themselves can act as both data producers and consumers, enabling progressive evolution without external supervision. However, despite their success in domains such as math and coding, the exploration of self-play or adversarial evolution remains largely absent in the context of tool use, leaving a critical research gap.

To address these challenges, we propose a novel **A**ntagonistic **E**volution method for **Tool** use (**AETool**). Unlike prior methods that directly rely on synthetic samples for training, AETool introduces a query-generation model and a tool-use model, which co-evolve through an adversarial pro-

---

[1]https://chatgpt.com
[2]https://www.anthropic.com

cess. The query-generation model rewrites queries based on tool-provided solutions in the dataset, thereby improving query quality while adaptively adjusting data complexity. The tool-use model is then trained on these rewritten samples to enhance its tool-use capabilities. Through adversarial optimization, the two models evolve iteratively: in each round, the query-generation model produces high-quality, diverse, and increasingly challenging samples informed by feedback from the tool-use model, which in turn is compelled to solve progressively harder problems. This iterative interaction fosters continuous refinement of both models, enabling the tool-use model to steadily improve its performance even under limited data conditions.

Our key contributions are summarized as follows:

- We propose AETool, the first adversarial evolution method tailored for tool-use tasks, enabling LLMs to enhance their tool-use ability without relying on massive synthetic datasets.
- We design a query-generation model that improves query quality and adaptively regulates task difficulty, addressing the limitations of low-quality or mismatched synthetic data in prior approaches.
- We establish an adversarial training loop between the query-generation model and the tool-use model, allowing both models to continuously refine their capabilities and achieve progressive improvement.
- Extensive experiments conducted on three benchmark tool-use datasets demonstrate that the model obtained by our AETool consistently outperforms state-of-the-art tool-use models, validating its effectiveness and robustnes.

## 2 RELATED WORK

### 2.1 TOOL-USE LLMS

A typical tool-use task consists of two core steps: selecting the most appropriate tool from a set of candidates and extracting the necessary parameters for tool invocation from the user's query. Existing research on this task can be broadly categorized into two approaches: non-tuning methods and tuning-based methods (Qu et al., 2025; Liu et al., 2023).

Non-tuning methods mainly rely on prompting strategies and few-shot learning. ReAct(Yao et al., 2023) models tool-use behavior explicitly by prompting the language model to "think and act" during the reasoning process. EasyTool(Yuan et al., 2025) proposes an automatic rewriting method to make tool descriptions more interpretable by the model. Concise(Xu et al., 2024b) summarizes tool functionalities using concise and clear language to reduce processing complexity while preserving semantic completeness. Another line of work(Shi et al., 2024) adopts a multi-agent collaboration strategy, thereby improving overall performance and task success rate.

Tuning-based methods, on the other hand, leverage tool-use samples to train existing LLMs, enabling them to systematically learn tool-use workflows and strengthen their invocation capabilities. These methods primarily focus on data collection and training strategies. Toolformer(Schick et al., 2024) introduces special tool-related tokens into the model's vocabulary to reformulate the language modeling task into a "call-response" structure, enabling explicit modeling of tool-use behavior during training. ToolkenGPT(Hao et al., 2024) builds upon this approach by introducing a multi-stage switching mechanism during decoding, allowing the model to dynamically alternate between text generation and tool invocation modes. Additionally, some studies(Qin et al., 2024; Yang et al., 2023; Liu et al., 2025) use advanced LLMs to automatically synthesize tool-use examples to enhance the capabilities of lightweight models through knowledge distillation.

### 2.2 SELF-EVOLVED LLMS

Self-evolution refers to the process by which a model gradually improves its capabilities through mechanisms such as self-learning, self-feedback, and self-optimization, without human intervention or external supervision. This process mainly involves experience acquisition, experience refinement, updating, and evaluation(Tao et al., 2024).

The earliest studies on self-evolution primarily focused on data construction and self-supervised fine-tuning. Self-Instruct(Wang et al., 2023) enables a language model to construct task instructions

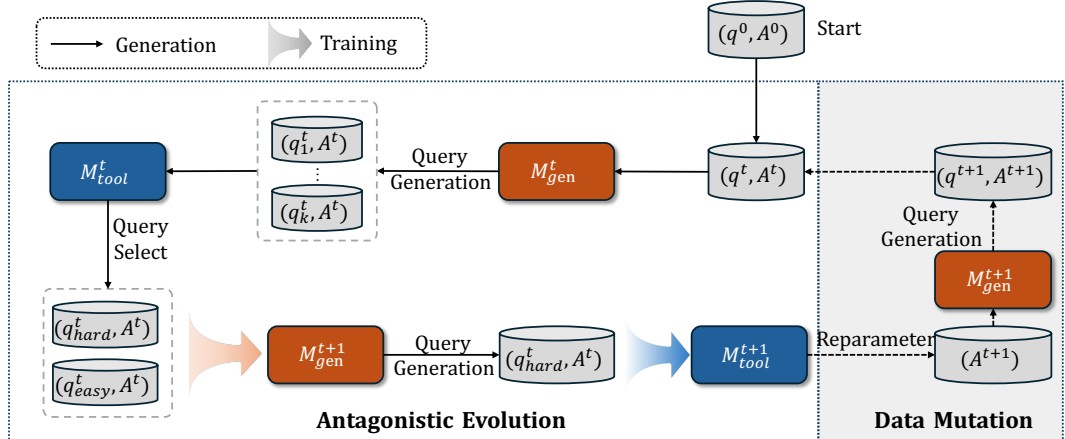

Figure 1: The overall workflow of the proposed antagonistic evolution, consisting of a query-generation model $\mathcal{M}_{gen}$ and a tool-use model $\mathcal{M}_{tool}$. In the left part, the two models are updated in an adversarial manner, where the generation model is optimized to generate more complex query and the tool model is optimized to achieve more tool-use accuracy. In the right part, mutation operations are conducted on training data to increase data diversity.

and train itself without human intervention. Building upon this, Evol-Instruct(Xu et al., 2024a) introduces a task evolution mechanism, allowing the model to achieve capability transitions through instruction sequences of increasing complexity.

Other kind of methods emphasize model reflection and self-correction. IterRefinement(Chen et al., 2024b) uses a series of prompts to encourage the model to improve upon its previous outputs; STaR(Zelikman et al., 2022) proposes guiding the model to perform rationale analysis and generate correct solutions based on identified patterns; Reflexion(Shinn et al., 2023) enables the model to iteratively revise and rewrite its generated content through self-evaluation, thus achieving multi-round optimization.

In recent years, some studies have introduced self-play mechanisms into the domain of large language models. SPIN(Chen et al., 2024c) treats the model's own generated answers as negative samples and applies preference optimization to update its parameters, achieving self-play dynamics. SPPO(Wu et al., 2025) incorporates evaluation metrics, allowing the model to generate multiple candidates, compare them, and update its parameters based on the best-performing responses, thereby driving strategy evolution.

## 3 METHODOLOGY

In this section, we introduce the proposed antagonistic evolution pipeline for tool-use, shown in Figure 1. The pipeline consists of a query-generation model and a tool-use model, where the query-generation model aims to produce high-quality samples and is optimized toward generating high-quality and increasingly complex training data. The algorithm is presented in Algorithm 1.

### 3.1 PRELIMINARIES

**Tool-use Task.** Given an LLM $\mathcal{M}$, a query $q$, and a set of candidate tools $\mathcal{T}$, the model is required to select the appropriate tools $t^i$ and fill in the parameters $(a_1^i, \cdots, a_m^i)$ to construct the final solution $A = [(t^i, a_1^i, \cdots, a_m^i), \cdots]$. This process is denoted as $(\mathcal{T}, q \xrightarrow{tool} A)$. Detailed prompts for this task are provided in Appendix B.

**Query-generation Task.** Given an LLM $\mathcal{M}$, a query $q$, a candidate tool set $\mathcal{T}$ and a solution $A$, the model is required to generate an improved query $q'$. The generated query should be more semantically complete, reasonable, and challenging than the original one. This process is denoted as $(\mathcal{T}, q, A \xrightarrow{gen} q')$. Detailed prompts for this task are provided in Appendix B.

---

**Algorithm 1:** Antagonistic Evolution

---

**Input:** $\mathcal{D} = \{ (\mathcal{T}_i, q_i^0 \xrightarrow{tool} A_i^0) \}_{i=1}^N$, a base LLM $\mathcal{M}$

1 , iteration number $T$. $\mathcal{D}_{tool}^0 = \mathcal{D}$;

2 $\mathcal{M}_{tool}^0 = \mathcal{M}$;

   /* Step 0:  Warmup query-generation model                                 */

3 $\mathcal{M}_{gen}^0 = WarmUp(\mathcal{D}, \mathcal{M})$;

4 **for** *t=0,...,T-1* **do**

      /* Step 1:  Updating query-generation model                     */

    5  $\mathcal{Q}^t = \text{QueryGeneration}(\mathcal{D}_{tool}^t, \mathcal{M}_{gen}^t, k)$ ;                  // Algo. 2

    6  $\mathcal{D}_{gen}^t = \{(\mathcal{T}_i, A_i^t, q_{i,easy}^t \xrightarrow{gen} q_{i,hard}^t)\}_{i=1}^N = \text{QuerySelect}(\mathcal{D}_{tool}^t, \mathcal{Q}^t, \mathcal{M}_{tool}^t)$ ;

      // Algo. 3

    7  $\mathcal{M}_{gen}^{t+1} = \arg\max_{\mathcal{M}} \frac{1}{N} \sum_{i=1}^N P_{\mathcal{M}}(q_{i,hard}^t \mid \mathcal{T}_i, A_i^t, q_{i,easy}^t)$ with $\mathcal{M}_{gen}^t$ as initialization;

      /* Step 2:  Updating tool-use model                                  */

    8  $\tilde{\mathcal{Q}}^t = \text{QueryGeneration}(\mathcal{D}_{tool}^t, \mathcal{M}_{gen}^{t+1}, 1)$ ;                  // Algo. 2

    9  $\tilde{\mathcal{D}}_{tool}^t = \{(\mathcal{T}_i, \tilde{\mathcal{Q}}_{i,1}^t \xrightarrow{tool} A_i^t)\}_{i=1}^N$;

    10  $\mathcal{M}_{tool}^{t+1} = \arg\max_{\mathcal{M}} \frac{1}{N} \sum_{i=1}^N P_{\mathcal{M}}(\tilde{q}_i^t \mid \mathcal{T}_i, A_i^t)$ with $\mathcal{M}_{tool}^t$ as initialization;

      /* Step 3:  Data Mutation                                           */

    11  $\mathcal{D}_{tool}^t = \text{Mutate}(\mathcal{D}_{tool}^t, \theta_{tool}^t)$ ;                           // Algo. 5

    12  $\mathcal{D}_{tool}^t = \text{QueryGeneration}(\mathcal{D}_{tool}^t, \theta_{gen}^t, 1)$;

13 **end**

**Output:** $\theta_{tool}^T$

---

## 3.2 WARMUP

A central component of our proposed evolution framework is the query-generation model, which iteratively rewrites queries to improve their quality and complexity. Since generating high-quality queries is inherently challenging and small-scale base models typically lack such advanced capability, we introduce a warm-up process prior to evolution in this section.

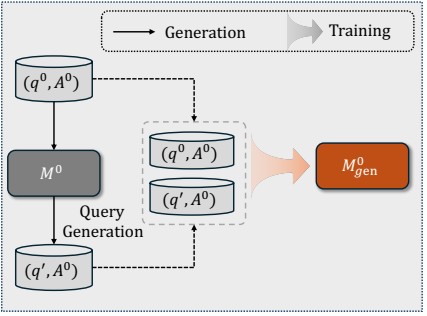

Figure 2: The warmup process for the query-generation model.

Given a tool invocation dataset $\mathcal{D} = \{ (\mathcal{T}_i, q_i \xrightarrow{tool} A_i) \}_{i=1}^n$, our goal is to optimize the base model for query-generation ability. First, the base model $\mathcal{M}$ is prompted to generate a query $q_i'$ based on the candidate tool sets $\mathcal{T}_i$ and solutions $A_i$. Since the base model has not been fine-tuned for the task, we assume it lacks the ability to produce high-quality queries. The generated queries typically fall short of real ones in terms of semantic completeness, plausibility, and informativeness, and are therefore treated as lower-quality counterparts to the original queries $\{q_i\}$. Second, we conduct supervised fine-tuning to the base model on the dataset $\{ (\mathcal{T}_i, A_i, q_i' \xrightarrow{gen} q_i) \}_{i=1}^n$, where the model is optimized to rewrite a more fluent, structurally sound and linguistically clear query from the lower-quality query. This significantly enhances the model's ability in query generation and query complication, offering an initial query-generation model $\mathcal{M}_{gen}^0$, which serves as a foundational component for the subsequent evolution process. The corresponding pipeline and algorithm are shown in Figure 2 and Algorithm 4 in Appendix A, respectively.

---

**Algorithm 2:** Query Generation

**Input:** $\mathcal{D}_{tool} = \{(\mathcal{T}_i, q_i \xrightarrow{tool} A_i)\}_{i=1}^N, \mathcal{M}_{gen}, k$

1   $\mathcal{Q} = \emptyset$;
2   **for** *i=1,...,N* **do**
3      $Q_i = \emptyset$;
4      **for** *j=1,...,k* **do**
5         $q'_{ij} \sim P(\cdot \mid \mathcal{T}_i, q_i, A_i, \mathcal{M}_{gen})$;
6         $Q_i = Q_i \cup \{q'_{ij}\}$;
7      **end**
8      $\mathcal{Q} = \mathcal{Q} \cup \{Q_i\}$;
9   **end**

**Output:** $\mathcal{Q}$

---

**Algorithm 3:** Query Select

**Input:** $\mathcal{D}_{tool} = \{(\mathcal{T}_i, q_i \xrightarrow{tool} A_i)\}_{i=1}^N, \mathcal{Q} = \{(q_{i,1}, \cdots, q_{i,k})\}_{i=1}^N, \mathcal{M}_{tool}$

1   $\mathcal{D}_{gen} = \emptyset$;
2   **for** *i=1,...,N* **do**
3      generate solutions $\tilde{\mathcal{A}}_i$ and confidence scores $P_{i,j}$ with $\mathcal{M}_{tool}$ for each query $q_{i,j}$ in $\mathcal{Q}_i$;
4      $\mathcal{Q}'_i = \{q_{i,j} \mid \tilde{\mathcal{A}}_{i,j} = A_i\}_{j=1}^k$;
5      $q_{i,easy} = \arg\max_{q_{i,j} \in \mathcal{Q}'_i} P_{\mathcal{M}_{tool}}(\tilde{\mathcal{A}}_{i,j} \mid \mathcal{T}_i, q_{i,j})$;
6      $q_{i,hard} = \arg\min_{q_{i,j} \in \mathcal{Q}'_i} P_{\mathcal{M}_{tool}}(\tilde{\mathcal{A}}_{i,j} \mid \mathcal{T}_i, q_{i,j})$;
7      $\mathcal{D}_{gen} = \mathcal{D}_{gen} \cup \{(\mathcal{T}_i, A_i, q_{i,easy} \xrightarrow{gen} q_{i,hard})\}$
8   **end**

**Output:** $\mathcal{D}_{gen}$

---

### 3.3 Antagonistic Evolution

In previous studies Qin et al. (2024); Liu et al. (2025); Lin et al. (2024), enhancing tool-use capabilities is typically achieved by directly performing supervised fine-tuning (SFT) on synthesized tool-instruction datasets. However, this approach imposes high demands on both the quantity and quality of the data, and it also faces a critical limitation: the complexity of the data must align with the model's capacity, otherwise the tool-use performance may remain suboptimal Liu et al. (2025). Inspired by the idea of Generative Adversarial Networks (GANs) Goodfellow et al. (2020), we train a query-generation model that shares the same origin as the tool-use model and employ it in an adversarial manner. The query-generation model $\mathcal{M}_{gen}$ and the tool-use model $\mathcal{M}_{tool}$ serve as the generator and discriminator in GANs, respectively, where $\mathcal{M}_{gen}$ is optimized to generate high-quality and increasingly challenging queries while $\mathcal{M}_{tool}$ is optimized to solve those challenging queries with correct tool-using solutions. This approach not only improves the quality of the training data but also enables adaptive adjustment to the complexity of the data.

In the evolution process, the two models are updated iteratively, resulting in the tool-use ability being increased through training on more challenging tasks. Additionally, a data mutation operation is conducted in each iteration to increase data diversity.

#### 3.3.1 Updating Query-Generation Model

Similar to the generative model in GANs, the optimization of the query-generation model $\mathcal{M}_{gen}$ aims to generate more challenging samples. The updating of the query-generation model $\mathcal{M}_{gen}$ comprises two crucial steps: data collection and model updating.

**Data collection.** For the previous iteration generated tool-invocation dataset $\mathcal{D}_{tool}^t = \{(\mathcal{T}_i, q_i^t \xrightarrow{tool} A_i^t)\}_{i=1}^N$, we use previous data generation model $\mathcal{M}_{gen}^t$ to generate $k$ candidate queries $\mathcal{Q}_i =$

$\{q_{i,j}\}_{j=1}^{k}$ for each $q_i$ with sampling method. Then we use the previous tool-use model to select an easy and a hard query, forming the training data set for the query-generation model. First, the tool-use model $\mathcal{M}_{tool}^t$ is instructed to generate solutions $A_i = \{A_{i,j}\}_{j=1}^{k}$ and confidence scores $P_i = \{p_{i,j}\}_{j=1}^{k}$ for each query. These confidence scores reflect the model's certainty in solving the query, serving as the complexity indicator of queries. However, the generation model may generate infeasible queries not be solved correctly, resulting in low-quality queries. To guarantee the quality of the selected queries, we select the two queries with the highest and lowest confidence scores from correctly-solved queries, representing the easy query $q_{i,\arg\max_j p_{i,j}}^t$ (denoted as $q_{i,easy}^t$) and $q_{i,\arg\min_j p_{i,j}}^t$ (denoted as $q_{i,hard}^t$) and the hard queries, respectively. Therefore, we obtain the query-paired data set $D_{gen}^t = \{(\mathcal{T}_i, A_i^t, q_{i,easy}^t \xrightarrow{gen} q_{i,hard}^t)\}_{i=1}^{N}$.

**Model updating.** Upon the query-paired samples being collected, the query-generation model $\mathcal{M}_{gen}^t$ is then optimized with the query-generation task: $(\mathcal{T}, A, q_{easy} \xrightarrow{gen} q_{hard})$ through a query-generation instruction, which can be formulated as:

$$\mathcal{M}_{gen}^{t+1} = \arg\max_{\mathcal{M}} \frac{1}{N} \sum_{i}^{N} P_{\mathcal{M}}(q_{i,hard}^t \mid \mathcal{T}_i, A_i^t, q_{i,easy}^t) \tag{1}$$

where the $\mathcal{M}$ is initialized with $\mathcal{M}_{gen}^t$. $P_{\mathcal{M}}(y \mid x)$ denotes the probability of generating $y$ for input $x$ of model $\mathcal{M}$, which is usually calculated by the next token prediction loss.

### 3.3.2 UPDATING TOOL-USE MODEL

Consistent with the update of the query-generation model, the update of the tool-use model $\mathcal{M}_{tool}$ consists of two steps: data complexification and model updating.

**Data complicating.** Following the update of the query-generation model, we obtain a model equipped with the ability to complexify queries. To further enhance the effectiveness of the tool-use model, we exploit this capability to perform an additional round of query complicating on the existing data, thereby generating more challenging training instances. Concretely, we employ the query-generation model to rewrite the queries in the previous dataset $\mathcal{D}_{tool}^t = \{(\mathcal{T}_i, q_i^t \xrightarrow{tool} A^t)\}_{i=1}^{N}$, and construct a new, more complex dataset $\tilde{\mathcal{D}}_{tool}^t = \{(\mathcal{T}_i, \tilde{q}_i^t \xrightarrow{tool} A^t)\}_{i=1}^{N}$. The resulting dataset serves as the training corpus for updating the tool-use model, thereby improving its ability to operate effectively under more complex query scenarios.

**Model updating.** The tool-use model $\mathcal{M}_{tool}^t$ is then trained on those new complex samples $\tilde{\mathcal{D}}_{tool}^t$ with the tool-use task: $(\mathcal{T}, q \xrightarrow{tool} A)$, which can be formulated as:

$$\mathcal{M}_{tool}^{t+1} = \arg\max_{\mathcal{M}} \frac{1}{N} \sum_{i}^{N} P_{\mathcal{M}}(\tilde{q}_i^t \mid \mathcal{T}_i, A_i^t) \tag{2}$$

where the $\mathcal{M}$ is initialized with $\mathcal{M}_{tool}^t$.

### 3.3.3 DATA MUTATION

Data diversity is essential to mitigate overfitting and enhance the robustness of the tool-use model. However, the candidate tools in each sample are unchanged in each iteration, introducing the risk that the tool-use model may exploit spurious correlations by memorizing a static mapping from candidate tools to solutions, thus ignoring the query content. Therefore, we introduce a reparameterization strategy to perturbs the solutions and then generate a novel query for the solutions. At the end of each iteration, the tool-use model perturbs the solution $A_i^t$ to a new one $A_i^{t+1}$. The new-generated solutions are validated via abstract syntax tree checks to ensure syntactic correctness. For each validated perturbed solution $A_i^{t+1}$, the data generation model $\mathcal{M}_{gen}^{t+1}$ produces a corresponding query $q_i^{t+1}$, yielding updated query-solution pairs $(q_i^{t+1}, A_i^{t+1})$. The final mutated dataset is thus defined as $\mathcal{D}_{tool}^{t+1} = \{(\mathcal{T}_i, q_i^{t+1} \xrightarrow{tool} A_i^{t+1})\}_{i=1}^{N}$.

Table 1: The performance of models on BFCL-live. SFT refers to model trained on the original dataset. **Bold** and underline(only Avg.) represent the best and the 2nd best results.

| Model | Simple | Multiple | Parallel | Parallel Multiple | Avg. |
|---|---|---|---|---|---|
| Llama3.1-8B-Instruct | 0.7674 | 0.7749 | **0.8750** | 0.7083 | 0.6108 |
| Qwen2.5-7B-Instruct | 0.7287 | 0.7569 | 0.6250 | 0.7083 | 0.7491 |
| Hammer2.1-7B | 0.7674 | 0.7740 | 0.8125 | 0.7083 | 0.7511 |
| Watt-tool-8B | 0.7674 | 0.7749 | **0.8750** | 0.7083 | 0.7650 |
| ToolACE-8B | 0.7326 | 0.7673 | 0.8125 | 0.7083 | 0.7602 |
| Falcon3-7B-Instruct | 0.7403 | 0.6648 | 0.7500 | 0.6250 | 0.5486 |
| BitAgent-8B | 0.7791 | 0.7740 | **0.8750** | 0.7083 | 0.7614 |
| SFT | 0.7868 | 0.7797 | 0.6875 | 0.6250 | 0.7772 |
| **Ours** | **0.8256** | **0.8025** | 0.8125 | **0.7917** | **0.8068** |

Table 2: The performance of models on API-Bank and ACEBench. The settings remain the same with the table above. *S-Turn*, *M-Turn* and *Similar* denote *Single Turn*, *Multiple Turn* and *Similar API* in ACEBench, respectively.

| Model | API-Bank | | | ACEBench | | | | |
|---|---|---|---|---|---|---|---|---|
| | Lv1 | Lv2 | **Avg.** | Atom | S-Turn | M-Turn | Similar | **Avg.** |
| Llama3.1-8B-Instruct | 0.7143 | 0.3852 | 0.5498 | 0.5100 | 0.4950 | 0.2800 | 0.6000 | 0.4740 |
| Qwen2.5-7B-Instruct | 0.7193 | 0.3259 | 0.5226 | 0.7000 | 0.5700 | 0.4900 | 0.6200 | 0.6310 |
| Hammer2.1-7B | 0.7494 | 0.4370 | 0.5932 | 0.7130 | 0.6250 | 0.4300 | 0.6400 | 0.6390 |
| Watt-8B | 0.7419 | 0.4519 | 0.5969 | 0.8470 | 0.7150 | 0.5700 | 0.7000 | 0.7590 |
| ToolACE-8B | 0.7218 | 0.3926 | 0.5572 | 0.8300 | 0.7300 | 0.5630 | **0.7840** | 0.7600 |
| Falcon3-7B-Instruct | 0.6667 | 0.3778 | 0.5223 | 0.6500 | 0.4800 | 0.4300 | 0.6800 | 0.5820 |
| BitAgent-8B | 0.7444 | 0.4370 | 0.5907 | 0.8470 | 0.7100 | 0.5500 | 0.7200 | 0.7450 |
| SFT | 0.7318 | 0.4889 | 0.6104 | 0.8130 | 0.7750 | 0.5600 | 0.7000 | 0.7560 |
| **Ours** | **0.7544** | **0.5037** | **0.6291** | **0.8500** | **0.8000** | **0.5800** | 0.6600 | **0.7750** |

## 4 EXPERIMENTS

### 4.1 EXPERIMENTAL SETTINGS

**Train Dataset and Base Model**. Our experiment utilizes 10,000 samples from xlam-function-calling-60k(Liu et al., 2024) dataset and the samples from ToolACE(Liu et al., 2025) dataset as training set, which is specifically designed for tool-use scenarios and contains structured function-calling samples. We also utilize Qwen2.5-7B-Instruct(Team, 2024a;b) as base model.

**Evaluation**. To validate the effectiveness of our method, we selected three different tool-using benchmarks for evaluation: BFCL(Yan et al., 2024), APIBank(Li et al., 2023) and ACEBench(Chen et al., 2025). Specifically, we used the BFCL-live as the primary benchmark for comparison, while the Level 1 and 2 evaluation sets from APIBank and the normal evaluation set from ACEBench served as complementary evaluations. This setup enables a more comprehensive assessment of the model's tool-use capabilities across various task types and domains.

**Baselines**. To validate the superiority of AETool over conventional training, we constructed multiple comparison baselines. First, we compared our approach with open-source models and fine-tuned tool-calling models of similar scale. Open-source models include Qwen2.5-7B-Instruct, LLaMA-3.1-8B-Instruct(AI@Meta, 2024) and Falcon3-7B-Instruct(Team, 2024c). Tool-calling models include Hammer2.1-7B (Lin et al., 2024), Watt-8B[3], ToolACE-8B(Liu et al., 2025) and BitAgent-8B[4].

---

[3]https://ollama.com

[4]https://bittensor.com/

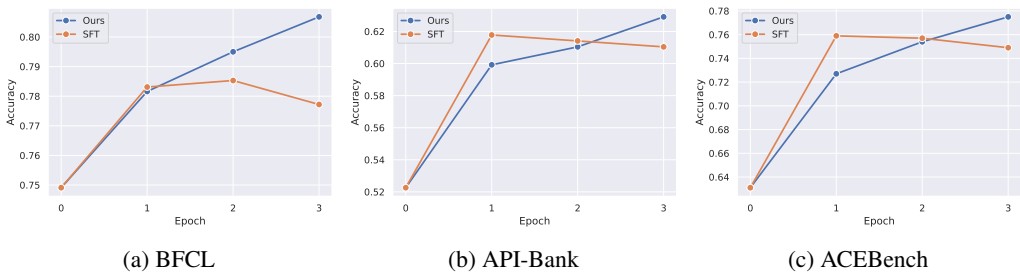

(a) BFCL       (b) API-Bank       (c) ACEBench

Figure 3: The performance of trained models over training iterations on three benchmarks.

Direct training on the original dataset without applying our evolution method is also included as a baseline, denoted as SFT in the results.

**Implementation Details**. Due to limited resources, all supervised fine-tuning in our method adopts a parameter-efficient tuning strategy LoRA (Hu et al., 2022). For hyperparameter settings, the rank is set to 8, alpha to 16, learning rate to $10^{-4}$, with a cosine learning rate scheduler and a warm-up ratio of 0.1. In the antagonistic evolution, the temperatures for query generation and tool use are set to 1 and 0.5, respectively. The number of new variants generated per query $k$ is set to 5.

## 4.2 MAIN RESULT

The overall evaluation results on the three benchmarks are illustrated in Table 1, Table 2 and Figure 3. We have the following findings according to the results:

*Findings 1:* Continuous training on the original dataset tends to cause overfitting and weaken tool-use performance. Further analysis reveals that performance degradation is observed across all three benchmarks. For all three benchmarks, performance increases during the first epoch but deteriorates steadily in the second and third epochs, exhibiting clear signs of overfitting. This suggests that the original training set may have overly concentrated or repetitive distributions and lack diversity. Prolonged training on such data leads the model to overfit specific patterns, which in turn undermines its ability to generalize and remain robust across broader tool-use tasks.

*Findings 2:* Our proposed method achieves state-of-the-art performance, consistently surpassing all baseline models across the evaluation benchmarks. Beyond the overall performance gains, the model demonstrates superiority in nearly all evaluation aspects within the three benchmarks, covering both general and fine-grained dimensions of tool-use capabilities. This comprehensive improvement highlights not only the effectiveness of our approach in handling diverse and complex tasks, but also its clear advantages over existing methods in enhancing tool-use capabilities.

*Findings 3:* Compared to conventional training methods, our approach exhibits robustness and yields consistently stable performance gains. Unlike the original models, our method shows consistent and significant performance improvements across all evaluation benchmarks. The immediate effectiveness further suggests that, unlike static training data, the samples generated by the model itself during tool-use interactions are closer to its capability boundaries. As a result, they are more targeted and adaptable, better stimulating and refining the model's potential. This not only enhances the model's ability to handle complex tasks but also highlights its capacity for self-evolution.

## 4.3 ABLATION STUDY

To validate the necessity and effectiveness of each component in our proposed method, we conducted ablation studies based on the theoretical framework of our approach. We designed three experimental settings as follows:

- **All**: The complete method as proposed in our paper.
- **All w/o Mutation**: Our method without the data mutation stage.
- **Alionl w/o UpdateGen**: Our method without updating query generation model stage.

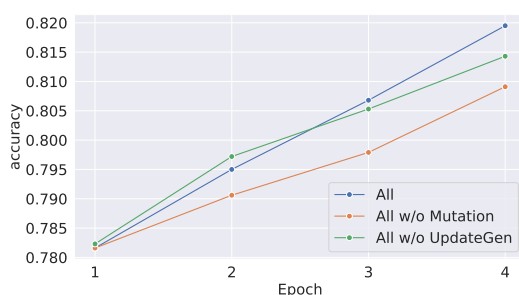

Figure 4: Ablation study on BFCL.

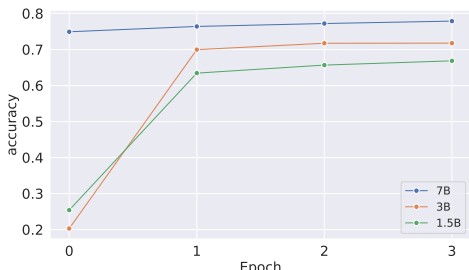

Figure 5: Scaling study on BFCL. (Base model: Qwen-2.5 series.)

We evaluated the models trained under these three configurations on the BFCL-live benchmark, with results shown in Figure 4. As illustrated, omitting the data mutation stage leads to a noticeable drop in performance, highlighting the importance of data diversity in enhancing model generalization. On the other hand, removing updating query generation model stage results in decent early-stage performance, but its improvement plateaus as training progresses, ultimately limiting further gains. In contrast, the complete method demonstrates a near-linear and stable performance improvement trend, confirming the long-term benefit of continuously introducing new data distributions during training. These comparisons strongly validate the necessity of each component in our system and further demonstrate the advantage of our method in sustaining performance gains.

## 4.4 SCALING STUDY

To evaluate the influence of model size on the performance of our method, we conducted a scaling study. Specifically, we applied our method to Qwen2.5-7B-Instruct, Qwen2.5-3B-Instruct and Qwen2.5-1.5B-Instruct, and evaluated the trained models on the BFCL-live benchmark. The experimental results are presented in Figure 5. As shown in the figure, our method consistently leads to performance improvements on all models. Notably, even with the smaller 3B and 1.5B models, we observe clear enhancements in tool-use capability after training. These findings further demonstrate the generality and scalability of our method, indicating its ability to effectively adapt to models of varying sizes while maintaining continuous performance gains.

## 4.5 DATA COMPLEXITY STUDY

To further validate whether the query-generation is optimized for generating increasingly challenging samples, we conducted evaluation on the generated samples in each iteration. Specifically, we evaluate Qwen2.5-7B-Instruct, Llama-3.1-8B-Instruct and Hammer2.1-7b on the generated data, presented in Table 3. It can be clearly seen that the accuracy of all

Table 3: The performance of models evaluated on data generated in 3 iterations in our evaluation process.

| Model | Iter 1 | Iter 2 | Iter 3 |
|---|---|---|---|
| Qwen2.5-7B-Inst | 0.7720 | 0.6712 | 0.6320 |
| Llama3.1-8B-Inst | 0.3938 | 0.3802 | 0.3638 |
| Hammer2.1-7B | 0.7403 | 0.7330 | 0.7210 |

models consistently decrease as the iteration increases, suggesting that the generated data is becoming challenging. Additionally, the deterioration is most pronounced for Qwen2.5-7B-Instruct, which serves as our base model, indicating the complexity of samples varies for different models, further verifying the mismatched complexity problem.

## 5 CONCLUSION

In this work, we propose AETool, an antagonistic evolution method for enhancing tool-use ability of LLMs, aimed at developing tool-use capabilities under the constraint of limited data quality and mismatched data difficulty. Extensive experiments demonstrate the effectiveness of our approach, highlighting its ability to enhance tool-use performance across diverse settings.

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

# A    ALGORITHM

We summarize the meaning of subscripts and superscripts used in the algorithms: the superscript $t$ denotes the t-th iteration, the subscripts $_{tool}$ and $_{gen}$ denote variables related to tool use and query generation, respectively, the subscripts $_{hard}$ and $_{easy}$ denote the hardest and easiest samples with the minimum and maximum confidence, respectively.

The rest of the algorithms, which were omitted from the main text for brevity, are presented below.

# B    TRAINING DETAILS

## B.1    QUERY GENERATION

**[SYSTEM]**
Here is a tool description: {tool_description}. The user will give you a conversation of a query

---

**Algorithm 4:** Warmup

---

**Input:** $\mathcal{D}_{tool} = \{(\mathcal{T}_i, q_i \xrightarrow{tool} A_i)\}_{i=1}^N)\}, \mathcal{M}_{gen}$

/* Generate unqualified queries */

1   $\mathcal{D}'_{gen} = \text{QueryGeneration}(\mathcal{D}_{tool}, \mathcal{M}_{gen}, 1)$ ;

2   $\mathcal{M}'_{gen} = \arg\max_{\mathcal{M}} \frac{1}{N} \sum_i^N P_{\mathcal{M}}(q'_i \mid \mathcal{T}_i, A_i^t, q_i)$ with $\mathcal{M}_{gen}$ as initialization;

**Output:** $\mathcal{M}'_{gen}$

---

**Algorithm 5:** Mutate

---

**Input:** $\mathcal{D}_{tool}, \mathcal{M}_{tool}$

1   $\mathcal{D}' = \emptyset$

2   **for** $i=1,...,N$ **do**

3      generate new answer $A'_i \sim P_{\mathcal{M}_{tool}}(\cdot \mid \mathcal{T}_i, A_i)$

4      **if** *ASTcheck($A'_i$)* **then**

5        $\mathcal{D}' \leftarrow \mathcal{D}' \cup \{(\mathcal{T}_i, q_i, A'_i)\}$

6      **else**

7        $\mathcal{D} \leftarrow \mathcal{D}' \cup \{(\mathcal{T}_i, q_i, A'_i)\}$

8      **end**

9   **end**

**Output:** $\mathcal{D}'$

---

and an answer. The query is unqualified either in providing enough value for the answer or in its reasonability. Based on the tool description and conversation, you should generate a better version of query. You should only output the query.

The query MUST follow the rules below: RULE1: The query MUST give out all the parameters in the answers. RULE2: Try to simulate and act as a normal human user asking a query in complete sentence. RULE3: The query MUST be in natural language.

**[USER]**
query:{query}
answer:{answer}

**[ASSISTANT]**
( generated query )

### B.2    TOOL USE

**[SYSTEM]**
Here is a set of tools: {tools}. The user will give you a query. Based on the tools and the query, you should generate the answer of the query. The query can be solved by one or more of the tools given. The answer should be in the format of [{{"name": function1_name, "arguments": {{param1_name: param1_value, param2...}}}}, function2...] NO other text MUST be included.

**[USER]**
{query}

**[ASSISTANT]**
( generated answer )

### B.3    MUTATION

**[SYSTEM]**
Here is a tool description: {tool_description}. The user will give you an answer. Based on the tool and answer, you should change the arguments' value of the answer. EVERY value of the generated answer should be different from the original one. The answer should be in the format of [{{"name": function1_name, "arguments": {{param1_name: param1_value, param2...}}}}, function2...] NO other text MUST be included.

**[USER]**
{answer}

**[ASSISTANT]**
( generated answer )

