# OpenReview forum: "Antagonistic Evolution for LLM Tool Use"
_ICLR.cc/2026/Conference — Submitted to ICLR 2026_

### Official Review · Reviewer_8C5p · 2025-10-17

**Soundness:** 2
**Presentation:** 3
**Contribution:** 2
**Rating:** 2
**Confidence:** 4

**Summary:**

This paper proposes an antagonistic evolution framework to train tool-use LLMs. The method iteratively optimizes the query-generation model that creates more complex queries, and the tool-use model that solves queries. Experiments on BFCL-live, APIBank, and ACEBench show performance improvements over baselines.

**Strengths:**

1. The paper tackles an important problem. Improving the tool-use ability of LLMs in a data-efficient and self-improving manner is highly relevant to current LLM research.
2. The co-evolution framework is an interesting and potentially effective idea. The design of alternating updates between query generation and tool-use training is conceptually reasonable.
3. The paper is overall clear and well written, easy to follow.

**Weaknesses:**

1. Although the paper claims to draw inspiration from GANs, the proposed antagonistic evolution does not involve an explicit adversarial optimization or equilibrium-seeking process.  There is no discussion of Nash equilibrium or convergence properties, and the algorithmic stability remains unclear.
2. The connection and distinction from curriculum learning, self-play, and self-evolution paradigms are not thoroughly discussed. The paper should clarify how the proposed method differs conceptually and technically from prior approaches.
3. Limited experiments:
* The experiments only use Qwen2.5-7B-Instruct as the base model, which limits generality.
* The baselines are mostly generic tool-learning models, but do not include methods like curriculum-learning or self-play-based methods for comparison.
* The discussion of results is insufficient. For instance, in Table 1, the proposed method underperforms several baselines under the “BFCL-live Parallel” setting, but this discrepancy is under explained.

**Questions:**

In Equation (2), the goal is to update the tool-use model. However, the current formulation maximizes the conditional probability of q̃ given A, rather than A given q̃. Is this correct？

---

### Official Review · Reviewer_QBxT · 2025-10-21

**Soundness:** 2
**Presentation:** 3
**Contribution:** 3
**Rating:** 2
**Confidence:** 4

**Summary:**

This paper proposes AETool, an antagonistic evolution framework to enhance the tool-use capability of large language models (LLMs). The key idea is to co-train a query-generation model and a tool-use model in an adversarial manner.. Extensive experiments across three benchmarks (BFCL, API-Bank, and ACEBench) demonstrate consistent performance gains over baselines.

**Strengths:**

1. Comprehensive Experiments. The paper evaluates on multiple public tool-use benchmarks.
2. Continuous Improvement. The results convincingly show that the model benefits from iterative training.

**Weaknesses:**

1. Overstated novelty. The paper claims to introduce the first adversarial evolution method for tool-use tasks. However, similar self-challenging or self-play paradigms have been explored in related agentic settings (e.g., [1–2]). These works are not cited or discussed, leaving unclear how AETool fundamentally differs from prior approaches.
2. Unfair comparisons. Many baselines are open-source models that differ in both backbones and training datasets. Consequently, it is difficult to determine whether the reported improvement originates from AETool itself, rather than from differences in model architecture or data scale.
3. Weak baseline selection. The only fair baseline is the vanilla SFT method, which is too weak to establish strong empirical evidence. The comparison omits recent reinforcement learning–based and self-challenging methods (e.g., [1–4]), limiting the strength of the empirical claims.
5. Lack of detailed analysis of the query model. The query model update appears to be the paper’s primary innovation relative to prior work [1]; therefore, its role warrants careful analysis, which is currently missing. Specifically, I have the following concerns:
	- Empirical issue: Figure 4 shows that removing the query-model update leads to only marginal degradation in later epochs and even performs better in early stages. This raises doubts about whether updating the query model is truly necessary.
	- Methodological issue: The paper trains the query model to generate harder queries based on the tool model’s confidence scores, but such confidence signals may be unreliable. Consequently, the distinction between “easy” and “hard” queries could be noisy, making it unclear whether the query model genuinely learns to generate more challenging queries.
	- Missing analysis: A more detailed investigation of the query model’s evolution and the actual difficulty growth of generated queries is needed to validate the claimed mechanism.


[1] Y. Zhou et al., Self-Challenging Language Model Agents

[2] P. Li et al., Iterative Tool Usage Exploration for Multimodal Agents via Step-wise Preference Tuning

[3] C. Qian et al., ToolRL: Reward is All Tool Learning Needs

[4] S. Zhang et al., Nemotron-Research-Tool-N1: Tool-Using Language Models with Reinforced Reasoning

**Questions:**

1. How sensitive is the performance to the temperature and the number of generated queries k in each iteration?
2. Does the adversarial process always converge to improvement, or can it degrade due to somehow unexpected queries?

---

### Official Review · Reviewer_m3db · 2025-10-23

**Soundness:** 2
**Presentation:** 3
**Contribution:** 2
**Rating:** 4
**Confidence:** 3

**Summary:**

This paper proposes AETool, an antagonistic co-evolution training scheme for tool use: a query-generation model iteratively rewrites the same solution into harder, higher-quality queries while a tool-use model learns to solve them; the authors report gains over similarly sized baselines and direct SFT on three mainstream tool-use benchmarks—BFCL, APIBank, and ACEBench.

**Strengths:**

Bringing antagonistic difficulty ramping into tool-use training is commendable; compared with the traditional pipeline that mass synthesizes data with a strong model and then SFTs it, this paper emphasizes self-generated samples aligned to the trained model’s competence frontier, which is a well-motivated direction consistent with recent self-play advances

**Weaknesses:**

1. Lack of theoretical analysis. First, this paper does not specify conditions under which antagonistic co-evolution avoids mode collapse and yields stable improvements. A minimal stylized objective plus stability conditions would help, such as self-play formulations with Nash-style guarantees. Secondly, there is no robustness analysis for the adversarially generated data: tool-using agents are known to be brittle under prompt-injection/perturbation and environment shifts, so a robustness section is needed. Thirdly, the method assumes the tool model’s confidence is a reliable proxy for difficulty, but LLM confidence is often miscalibrated.
---
2. Related work may be incomplete. This paper should more systematically compare with prior adversarial/self-evolution or self-generated data approaches for tool use and alignment.

**Questions:**

APIBank Level-3 exists and is widely regarded as a key tier for integrated multi-tool/multi-turn planning; please clarify why it was not evaluated or include a Level-3 assessment to strengthen completeness.

---

### Official Review · Reviewer_6D4F · 2025-11-01

**Soundness:** 3
**Presentation:** 3
**Contribution:** 3
**Rating:** 6
**Confidence:** 3

**Summary:**

The paper proposes AETool, a novel antagonistic evolution framework for improving LLM tool-use capabilities through co-evolution of a query-generation model and a tool-use model. Unlike most self-evolution approaches that rely on a single model, AETool introduces an adversarial training loop with data mutation to iteratively generate more challenging yet solvable queries. The method achieves state-of-the-art results across three major tool-use benchmarks.

**Strengths:**

1.Most prior self-evolving approaches rely on a single model to generate and learn from its own data. In contrast, AETool explicitly decouples roles into a query-generation model and a tool-use model, enabling more controlled and adaptive difficulty progression.
2.The proposed system integrates four key components: query generation, tool use, adversarial co-evolution, and data mutation into a cohesive and iterative pipeline that enables continuous mutual improvement.
3.AETool consistently achieves state-of-the-art results on three established tool-use benchmarks, demonstrating both effectiveness and generalization.

**Weaknesses:**

1.Each iteration requires running two models, multiple sampling passes, and a data mutation step, leading to significantly higher training costs than standard supervised fine-tuning (SFT). The paper lacks key efficiency metrics (e.g., training time, GPU hours, or inference latency), making it difficult to assess practical deployability.
2.While the method is described as antagonistic or GAN-inspired, it could alternatively be interpreted as a form of multi-agent collaborative training. The paper does not clearly differentiate its approach from existing multi-agent frameworks or justify why the adversarial perspective is more appropriate.
3.The warmup phase for the query-generation model depends on high-quality ground-truth queries from the original dataset. If the initial data is noisy or low-quality, the entire evolution process may start from a poor initialization. The robustness of the method under such conditions is not analyzed.

**Questions:**

Please see weaknesses

---

### Meta-Review · Area_Chair_8ebP · 2026-01-07

**Summary:**

Reviewers find the co-evolution idea promising and results competitive on several tool-use benchmarks, but the overall consensus trends negative due to concerns about overstated novelty, unclear “adversarial/GAN” framing and stability, missing/weak or potentially unfair baselines, limited analysis (especially of the query model’s necessity and difficulty growth), and lack of efficiency/robustness evaluation.

**Reviewer Concerns:**

With no rebuttal, key concerns remain.

**Reviewer Scores:**

With no rebuttal, key concerns remain.

---

### Decision · Program_Chairs · 2026-01-26

Reject